# Risk Factors Associated with Mechanical Ventilation in Critical Bronchiolitis

**DOI:** 10.3390/children8111035

**Published:** 2021-11-11

**Authors:** Rachel K. Marlow, Sydney Brouillette, Vannessa Williams, Ariann Lenihan, Nichole Nemec, Joseph D. Lukowski, Cheng Zheng, Melissa L. Cullimore, Sidharth Mahapatra

**Affiliations:** 1Department of Pediatrics, University of Nebraska Medical Center, Omaha, NE 68198, USA; rmarlow@childrensomaha.org (R.K.M.); mcullimore@childrensomaha.org (M.L.C.); 2Children’s Hospital & Medical Center, Omaha, NE 68114, USA; sbrouillette@childrensomaha.org (S.B.); vwilliams@childrensomaha.org (V.W.); alenihan@childrensomaha.org (A.L.); 3Boys Town National Research Hospital, Omaha, NE 68010, USA; nichole.nemec@boystown.org; 4Department of Neuroscience, The University of Nebraska-Omaha, Omaha, NE 68182, USA; jdlukowski87@gmail.com; 5Department of Biostatistics, University of Nebraska Medical Center, Omaha, NE 68198, USA; cheng.zheng@unmc.edu

**Keywords:** acute respiratory failure, bacterial pneumonia, critical bronchiolitis, invasive mechanical ventilation, non-invasive respiratory support

## Abstract

The American Academy of Pediatrics (AAP) recommends supportive care for the management of bronchiolitis. However, patients admitted to the intensive care unit with severe (critical) bronchiolitis define a unique group with varying needs for both non-invasive and invasive respiratory support. Currently, no guidance exists to help clinicians discern who will progress to invasive mechanical support. Here, we sought to identify key clinical features that distinguish pediatric patients with critical bronchiolitis requiring invasive mechanical ventilation from those that did not. We conducted a retrospective cohort study at a tertiary pediatric medical center. Children ≤2 years old admitted to the pediatric intensive care unit (PICU) from January 2015 to December 2019 with acute bronchiolitis were studied. Patients were divided into non-invasive respiratory support (NRS) and invasive mechanical ventilation (IMV) groups; the IMV group was further subdivided depending on timing of intubation relative to PICU admission. Of the 573 qualifying patients, 133 (23%) required invasive mechanical ventilation. Median age and weight were lower in the IMV group, while incidence of prematurity and pre-existing neurologic or genetic conditions were higher compared to the NRS group. Multi-microbial pneumonias were diagnosed more commonly in the IMV group, in turn associated with higher severity of illness scores, longer PICU lengths of stay, and more antibiotic usage. Within the IMV group, those intubated earlier had a shorter duration of mechanical ventilation and PICU length of stay, associated with lower pathogen load and, in turn, shorter antibiotic duration. Taken together, our data reveal that critically ill patients with bronchiolitis who require mechanical ventilation possess high risk features, including younger age, history of prematurity, neurologic or genetic co-morbidities, and a propensity for multi-microbial infections.

## 1. Introduction

Although viral bronchiolitis is usually a self-limited affliction characterized by low-grade fever, congestion, and rhinorrhea, severity of symptoms can be both variable and unpredictable, resulting in a 2–3% admission rate amongst infants, making it the most common cause of hospitalization in the first year of life [1,2,3,4,5]. Despite the wide spectrum of etiologic causes, respiratory syncytial virus (RSV), which accounts for up to 80% of cases, has been linked with more severe disease compared to non-RSV bronchiolitis, especially in the premature population [1,3,6,7,8].

The American Academy of Pediatrics (AAP) recommends supportive care alone and against the routine use of chest radiography, routine laboratory investigations, bronchodilators, hypertonic saline, or systemic steroids in the management of acute bronchiolitis [8]. However, no guidelines exist for the management of critical bronchiolitis requiring admission to the pediatric intensive care unit (PICU), which represents 15–22% of admitted bronchiolitis patients [5,9,10]. Distinguishing factors predictive of higher illness severity include demographic features (low birth weight [11], prematurity [2,7], young age [2,5,7,12]), clinical features at admission (tachypnea [11], retractions [5,12], or desaturation at admission [5]), and comorbidities (chronic lung disease, congenital heart disease, Trisomy 21, or neuromuscular disorders [2,3,7]). However, none of these studies identified risk factors associated specifically with higher likelihood of invasive mechanical ventilation.

Non-invasive respiratory support (NRS) systems, including heated high-flow nasal cannula (HHFNC), continuous positive airway pressure (CPAP), bilevel positive airway pressure (BiPAP), and RAM cannula, are commonly used to address acute respiratory failure, both in the acute care and critical care settings. Although RAM cannula is approved as a class 1 oxygen delivery device, proper fitting prongs that occupy 60–70% of the nares can deliver positive pressure in neonates and infants [13]. Randomized trials in adults have shown a reduction in intubation rates with the use of NRS for acute respiratory failure, which has translated into practice guidelines by the European Respiratory Society and the American Thoracic Society [14,15,16]. Similarly in pediatrics, NRS has shown success in decreasing the need for invasive mechanical ventilation (IMV) [17,18]. In critical bronchiolitis, HHFNC alone was shown to dramatically reduce intubation rates [19,20,21]. Thus, NRS has become a favored mode of treating respiratory failure with hypoxia and/or hypercarbia secondary to critical bronchiolitis [22,23,24,25].

Even with NRS support, a subset of patients with bronchiolitis tend to worsen over their disease course and eventually require mechanical ventilation, representing between 2–10% of all bronchiolitis admissions. Amongst critically-ill children, intubation rates vary from 10–15% for RSV bronchiolitis to 25% for non-RSV bronchiolitis [9,10]. In assessing responsiveness to NRS, a single-site study in a small cohort of patients identified an FiO_2_ of >0.8 for up to 60 min as a criterion for failure [26]. However, no other guidelines or clinical characteristics have been identified to distinguish pediatric patients with acute respiratory failure who may need invasive mechanical ventilation to adequately address their oxygenation and ventilatory needs.

In this study, we examined patients with acute respiratory failure in the setting of critical bronchiolitis managed on NRS to identify clinical and demographic features that differentiated the NRS group from the IMV group. We also compared patients intubated early in their disease course with those intubated later to determine if there were inherent differences between these groups. With our findings, we produced a predictive model using selected demographic and clinical data that may prove helpful in early identification of pediatric patients with critical bronchiolitis likely to require mechanical ventilation.

## 2. Materials and Methods

### 2.1. Setting

The Children’s Hospital & Medical Center (CHMC), Omaha, is a 145-bed tertiary pediatric medical center and the only free-standing pediatric hospital in Nebraska. CHMC houses a 32-bed combined cardiac/non-cardiac pediatric intensive care unit with an annual admission rate of approximately 1100 patients, an average daily census of 21, and a standardized mortality ratio of 0.87.

### 2.2. Study Design

Our study received approval from the Institutional Review Board (IRB) of the University of Nebraska Medical Center (UNMC) as a minimal risk study with a waiver of informed consent (Protocol: 655-17-EP). We adhered strictly to the ethical principles outlined in the Declaration of Helsinki (2013) and were HIPAA compliant. For this retrospective cohort study, the electronic medical record (EMR) was interrogated for all pediatric patients admitted to the PICU with a diagnosis of acute bronchiolitis from January 2015 to December 2019.

### 2.3. Eligibility

For inclusion in the study, patients had to be: (1) ≥37 weeks corrected gestational age, older than 72 h, and ≤2 years old, (2) carry an ICD-9 or ICD-10 diagnosis of acute bronchiolitis (refer to Appendix B), and (3) be managed on NRS excluding HHFNC, i.e., CPAP, BiPAP, and/or RAM cannula, or IMV. Patients were excluded for: (1) never requiring higher NRS than HHFNC during their PICU stay, (2) baseline chronic ventilatory support, (3) congenital heart disease with single ventricle physiology, and (4) immediate post-operative status.

### 2.4. Variables

Demographic and historical information included age and weight at PICU admission, gender, gestational age at delivery, any pre-existing neurologic or genetic conditions, self-reported race and ethnicity, and insurance type (as a surrogate for socioeconomic status). Hospital course information included duration of intubation, ventilator-free-days, PICU length of stay (LOS), severity of illness based on the pediatric index of mortality-III risk of mortality (PIM-III ROM) score [27], vasoactive medication use, and in-hospital mortality. Data related to infecting pathogens included number and type of infecting pathogens, white blood cell (WBC) count closest to the time of intubation, percent bands closest to the time of intubation, procalcitonin closest to the time of intubation, use and duration of antibiotics, and timing of antibiotic initiation relative to intubation. Pathogen positivity in the NRS group was based on respiratory viral panel testing (turnaround ~60 min) at the time of admission, which tests for the following pathogens: adenovirus, coronavirus (229E, HKU1, NL63, OC43), human metapneumovirus, rhinovirus/enterovirus, influenza A and B, parainfluenza 1–4, respiratory syncytial virus, *Bordetella pertussis*, *Chlamydophila pneumoniae*, and *Mycoplasma pneumoniae*. In the IMV group, tracheal aspirates were obtained at the time of intubation or admission (if transferred intubated) and sent for respiratory culture and gram stain.

### 2.5. Definitions

Critical bronchiolitis was defined as any acute bronchiolitis diagnosis necessitating management in the PICU for risk of impending respiratory failure. At our institution, transfer to the PICU is usually triggered when a child’s respiratory support exceeds 2 mL/kg of flow through a high-flow nasal cannula delivery system. Acute respiratory failure was defined as needing non-invasive respiratory support to maintain oxygen saturation ≥ 88% and/or to address work of breathing. Non-invasive respiratory support (NRS) included CPAP, BiPAP, and RAM cannula; HHFNC was excluded since many patients at our center on HHFNC are managed on the medical/surgical floor, like many other centers [5,28,29], and thus would not meet a diagnosis of critical bronchiolitis. We deliver NRS via RAM cannula by assigning a peak inspiratory pressure (PIP), peak end-expiratory pressure (PEEP), respiratory rate, inspiratory time (i-time), and fraction of inspired oxygen (FiO_2_), using a conventional mechanical ventilator for delivery. The nasal prongs are fitted to occupy ≥60% of the diameter of the patient’s nares in order to approximate delivery of positive pressure [13]. CPAP, continuous positive airway pressure, and BiPAP, bilevel positive airway pressure, are provided through a conventional mechanical ventilator via nasal or face mask. Any patient requiring invasive mechanical ventilation was placed in the “IMV” group; within the IMV group, patients intubated within 24 h of admission were placed in the “early IMV” group, while those intubated greater than 24 h after admission were placed in the “late IMV” group. The early IMV group was further subdivided into patients who arrived intubated vs. those who progressed to invasive mechanical ventilation after admission. Ventilator-free days were defined as the number of days alive and off the ventilator 28 days following intubation [30].

### 2.6. Statistical Analysis

All continuous variables are presented with medians and interquartile ranges or mean and standard deviation, while categorical variables are presented using frequencies and percentages. The chi-square test of independence was used to compare categorical data, and the Wilcoxon rank-sum test was used when comparing continuous variables, in case of normal distribution failure. Multivariate logistic regression models were built to predict the need for intubation. Stepwise selection with entry and stay *p*-value threshold set at 0.1 were used to select the variables that most strongly associated with the need for intubation. Receiver operating curve (ROC) for the prediction model was computed, and the area under the curve (AUC) was used to summarize the overall predictive power of the model. Statistical significance was established at *p* < 0.05.

## 3. Results

We identified a total of 775 patients on EMR interrogation with a diagnosis of acute bronchiolitis. After excluding 202 patients based on established criteria, of the remaining 573 eligible patients, 133 (23%) required IMV, while 440 (77%) were managed non-invasively. Upon subgroup analysis of the IMV group, 96 patients (17%) were identified as early IMV and 37 (6%) were intubated later in their PICU stay. Further subdividing patients in the early IMV group, 52 patients (9%) were intubated prior to arrival, while the rest (8%) progressed to invasive mechanical ventilation within 24 h of admission (Figure 1).

When examining demographic data between all groups, there were no differences in sex, race, ethnicity, or insurance type. However, the NRS and IMV groups differed significantly in age (5 months vs. 3 months, *p* = 0.002) and weight (7.0 kg vs. 5.3 kg, *p* < 0.001); within the IMV group, these differences were neither discernable between early and late intubation (Table 1) nor between those intubated prior to or after admission (Appendix A).

In comparing clinical characteristics between the NRS and IMV groups, differences were noted in gestational age at delivery, pre-term status at birth, and presence of pre-existing genetic or neurologic conditions. More specifically, the IMV group had a significantly younger median gestational age at delivery (37 weeks vs. 38 weeks, *p <* 0.001); had a higher proportion of patients with a history of prematurity (*p <* 0.001); and a higher incidence of pre-existing genetic (12% vs. 5%, *p* = 0.0042) and neurologic (20% vs. 6%, *p <* 0.001) conditions. Within the IMV group, a higher incidence of pre-existing genetic conditions was noted in the late IMV group (22% vs. 8%, *p* = 0.035) (Table 2). These differences were not discerned amongst patients intubated prior to versus after arrival to the PICU (Appendix A).

Next, we compared hospital course between groups. Expectedly, the IMV group experienced a longer PICU length of stay (8 days vs. 2 days, *p <* 0.001) and higher severity of illness (1.1% vs. 1.0%, *p =* 0.007), as evidenced by vasoactive drug usage (12% vs. 0%, *p <* 0.001) and overall in-hospital mortality (1.5% vs. 0%, *p =* 0.01). Similarly, the late IMV group experienced a longer ICU length of stay (10 days vs. 7 days, *p <* 0.001) and more vasoactive medication usage (24% vs. 7%, *p =* 0.007), contrarily with a lower severity of illness (1% vs. 1.1%, *p =* 0.008) and no differences in mortality from the early IMV group (Table 3). The late IMV group also had a longer during of intubation (7 days vs. 6 days, *p =* 0.043) and in turn, less ventilator-free days (21 days vs. 22 days, *p =* 0.041) (Table 3). Within the early IMV group, patients intubated after admission experienced a longer duration of intubation (7 days vs. 5 days, *p =* 0.002) and, in turn, less ventilator-free days (21 days vs. 23 days, *p =* 0.002), associated with longer PICU LOS (8 days vs. 6 days, *p =* 0.005) but not higher illness severity, vasoactive usage, or mortality (Appendix A).

Delving into the pathogen characteristics between groups, several interesting observations were highlighted. While there were no differences in the pathogen positivity between groups, the burden was significantly higher in the intubated cohort with a median difference of 1 pathogen (*p <* 0.001). Furthermore, the NRS group was significantly more likely to have single pathogens (75%) and only viral pathogens identified (98%), while ≥70% of the IMV group had evidence of multi-microbial or mixed (virus + bacterial) infections (*p <* 0.001). The only discernible difference within the IMV group was the presence of a higher pathogen load in the late IMV group (3 vs. 2 pathogens, *p =* 0.007) (Table 3 and Appendix A).

As noted in multiple prior studies [2,31,32], RSV and rhino/enterovirus were the top causes of bronchiolitis in all patients; *H.influenzae*, *M.catarrhalis*, and *S.pneumoniae* were the most frequently isolated bacterial pathogens in all IMV groups (Appendix A). In comparing RSV vs. non-RSV infections, overall younger patients were afflicted with RSV vs. non-RSV viruses. Moreover, patients with RSV bronchiolitis experienced longer hospitalizations compared to non-RSV bronchiolitis in the NRS group, which corroborates prior published observations [7,8]; this difference was lost in the IMV group (Appendix A).

Finally, when examining antibiotic usage, a significantly higher proportion of patients in the IMV group received antibiotics and for a longer duration than the NRS group (*p <* 0.001). Although the median duration of antibiotics was 7 days in both groups, the NRS group used antibiotics for an average of 6.8 days, while the IMV group used antibiotics for an average of 9.5 days (*p* < 0.001). We also found that those in the late IMV subgroup received longer courses of antibiotics with a median difference of 3 days (*p =* 0.017) (Table 4). Within the early IMV group, those intubated after arrival experienced later initiation of antibiotics (7 h vs. 0 h, *p <* 0.001) and longer duration of antibiotics (10 days vs. 7 days, *p =* 0.007) (Appendix A). No discernible differences were observed in lab findings amongst groups.

Using our data, we generated a model for determining the probability of intubation in critical bronchitis patients. The most important variables included weight, gestational age ≤ 36 weeks, and presence of a pre-existing neurologic or genetic condition. The latter was found to have the highest odds ratio in our model (Figure 2A). The area under the receiver operating characteristic curve of 0.72 was acceptable [33] for determining whether mechanical ventilation is likely based on the chosen variables (Figure 2B).

We separately conducted the same analysis including pathogen data, which took into account both bacterial and viral pathogen positivity and burden. This dramatically improved the AUC to 0.86. However, in the clinical context, we typically identify bacterial pathogens after intubation. Hence, its practical utility in a predictive model is limited. That said, this analysis did highlight the important role multi-microbial infections likely play in the need for invasive mechanical support (Appendix A).

## 4. Discussion

As the most common cause of hospitalization in infants, bronchiolitis poses a large burden on our healthcare system [1,3,5,34]. Critical bronchiolitis represents a unique subgroup of patients with a distinct disease trajectory punctuated by the need for intensive care in up to a quarter of admitted cases; between 10–25% of these patients will go on to require invasive mechanical ventilation [5,9,10]. However, clinical characteristics that distinguish the IMV group from their NRS counterparts are lacking. By identifying risk factors that are independently associated with the need for mechanical ventilation in critical bronchiolitis, intensivists may better manage these patients with targeted approaches.

We observed a 23% incidence of mechanical ventilation in critically ill bronchiolitis patients admitted to our unit; removing patients intubated prior to arrival, our observed intubation rate for patients managed on NRS and progressing to IMV is 14%, similar to prior studies [9,10]. Almost ¾ of patients in the IMV group were intubated within 24 h of presentation; and, for those intubated early, nearly half were first managed on NRS and then progressed to requiring invasive mechanical support. Expectedly, intubated patients had more complicated hospitalizations with longer lengths of stay, higher severity of illnesses and mortality. We were able to hone in on risk factors at admission that may help identify these children early, with the strongest predictors for IMV being weight, which relates to age at admission, history of prematurity, and having a pre-existing genetic or neurologic condition. Our data are corroborated by prior studies examining RSV bronchiolitis in infants, identifying risk factors that portend more complicated hospitalizations, including prematurity, low birthweight, young age, and co-morbidities [1,2,3,5,7,11,12,35]. Our study adds new insight by not only highlighting the contribution of the same risk factors but also the role of multi-microbial infections in the eventual need for invasive mechanical ventilation amongst critically ill bronchiolitis patients.

A potentially discerning aspect in the need for IMV in critical bronchiolitis may arise from complicating bacterial pneumonias. The AAP discourages the routine use of antibiotics in this population unless there is a clear source of bacterial infection [8,36]. In intubated critical bronchiolitis patients, despite practice variations across the country, early antibiotic initiation was associated with shorter duration of mechanical ventilation and shorter hospitalizations [37]. However, this study did not identify etiologic bacterial pathogens in these patients. Within our IMV cohort, 72% had evidence of a bacterial infection. Moreover, these patients had a higher pathogen load and multi-microbial infections compared to the NRS group. Of note, at our institution, we only obtain tracheal aspirates for culture in intubated patients; in addition, our respiratory PCR panel for nasopharyngeal samples checks for a limited cadre of bacterial pathogens, including *Mycoplasma pneumoniae*, *Chlamydia pneumoniae*, and *Bordetella pertussis* and *parapertussis*. Despite these limitations, it is noteworthy that a minority of unintubated patients received antibiotics (29% in NRS vs. 94% in IMV), yet experienced shorter PICU stays and better illness severity compared to those that were intubated. In fact, antibiotic usage in the NRS group had no effect on PICU length of stay. Taken together, these data associate invasive mechanical ventilation with bacterial coinfections in critical bronchiolitis. While data are not present to conduct intergroup comparisons on bacterial pathogen load between the NRS and IMV groups, the lower antibiotic usage and less severe hospitalization in the former would support this association.

What role does timing of intubation play in disease trajectory? Based on our data, delayed intubations were associated with more severe hospitalizations, longer PICU LOS, higher vasoactive usage, and longer duration of antibiotics. Our seemingly divergent finding of lower severity of illness in the late IMV group reflects similar findings by Kopp et al. who noted that patients on NRS for ≥24 h before progressing to IMV had a lower risk of mortality but longer duration of IMV than their age-matched cohorts who did not receive NRS prior to IMV [38]. Of note, ≥50% of patients in our early IMV group were intubated prior to arrival to the PICU. As we are a referral center for a large catchment area of rural hospitals that may not have a full range of NRS options for pediatric patients, we cannot ascertain the proportion of patients in the early IMV group who truly needed IMV and who could have instead benefitted from NRS. Moreover, whether physicians in rural settings were influenced to intubate patients based on factors within our predictive model remains unknown.

As a single center retrospective study, we were constrained by the information recorded in the EMR. For example, the influence of duration of illness prior to admission and history of palivizumab administration are important variables that were not accurately reflected. Moreover, given the high proportion of *Hemophilus influenzae* and *Streptococcus pneumoniae* in the IMV group and their young median age, vaccination status would have been an interesting additional variable to examine. Moreover, risk factors highlighted by prior studies for NRS failure have included FiO_2_ requirements, respiratory rate changes following NRS initiation, and ventilation/perfusion mismatch [18,26]. While not examined in our study, these factors should be considered in prospective trials. Finally, the prediction model we generated would need validation in a prospective clinical trial, and currently stands to simply highlight the risk factors identified by this study on which patients with critical bronchiolitis would potentially require mechanical ventilation.

## 5. Conclusions

Within the critical bronchiolitis cohort of pediatric patients, those requiring invasive mechanical ventilation are clinically distinct from those managed non-invasively. These patients are typically younger at admission, have a history of prematurity, and can have pre-existing neurologic or genetic co-morbidities. Their need for mechanical ventilation may be related to bacterial co-infections, potentially leading to complicated multi-microbial pneumonias, that can worsen disease trajectory and prolong hospitalization. Future studies should focus on determining early signs of deterioration which may be acted upon to potentially reduce the need for invasive ventilation and, in turn, improve outcomes.

## Figures and Tables

**Figure 1 children-08-01035-f001:**
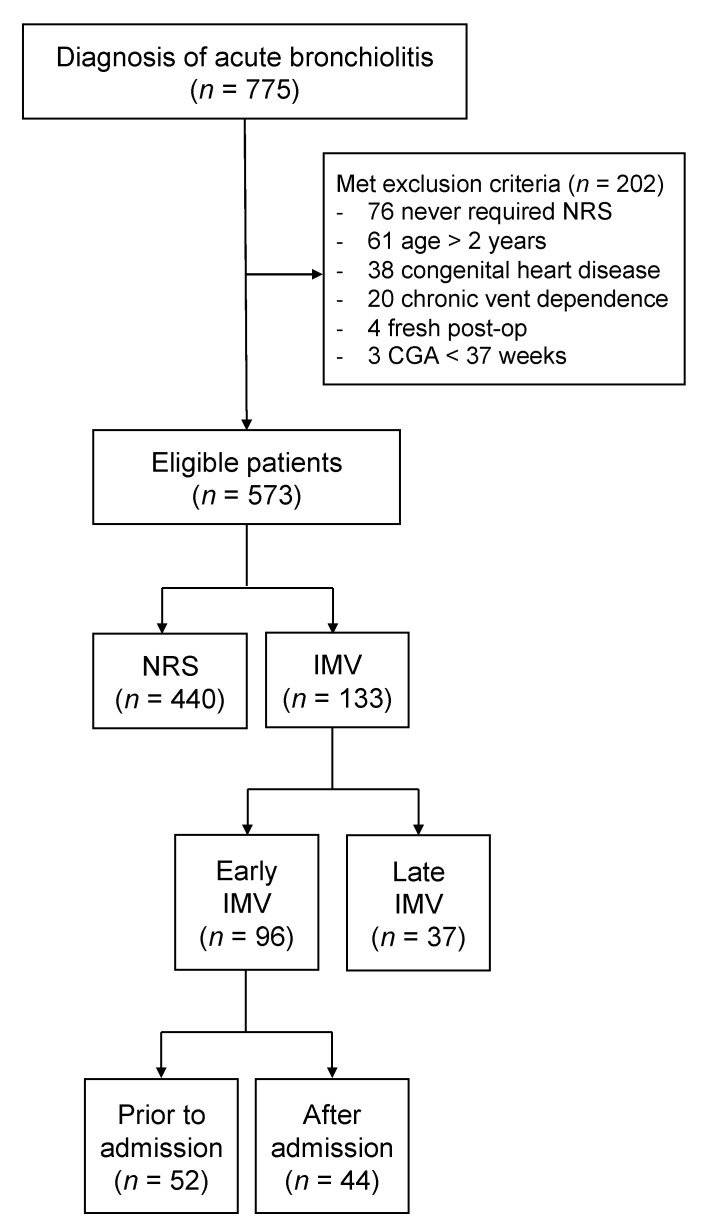
Patient enrollment flow diagram. CGA = corrected gestational age, IMV = invasive mechanical ventilation, NRS = non-invasive respiratory support, post-op = post-operation, vent = ventilator.

**Figure 2 children-08-01035-f002:**
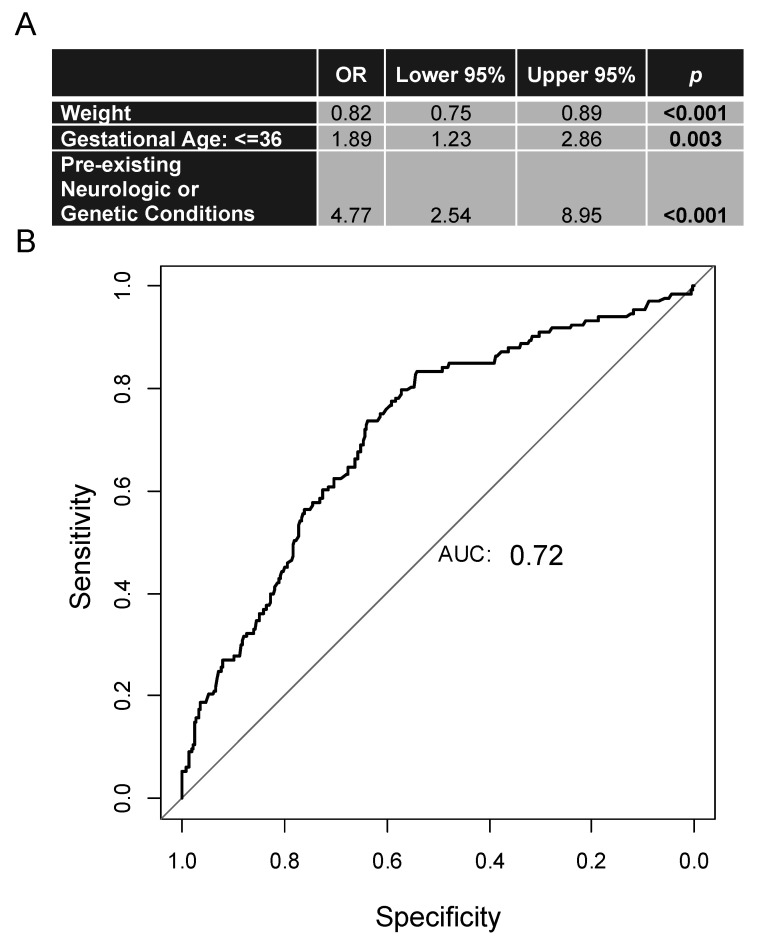
Predictive model for critical bronchiolitis patients requiring invasive mechanical ventilation. (**A**) Table of variables used to generate the predictive model. (**B**) Receiver operating characteristic (ROC) curve taking into account clinical characteristics with high odds of requiring mechanical ventilation. AUC = area under the curve.

**Table 1 children-08-01035-t001:** Patient Demographics (all groups, *n* = 573).

	NRS (440)	IMV (133)	*p* ^†^	Early IMV (96)	Late IMV (37)	*p* ^†^
Age at PICU admission						
Median (IQR), mo	5 (2–11)	3 (1.7–6.5)	**0.002**	3 (1.5–6)	3 (2–7.5)	0.23
*n* (%)						
≤1 month	35 (8)	19 (14)	**0.022**	16 (17)	3 (8)	0.39
2–12 months	315 (72)	97 (73)		69 (72)	28 (76)	
13–24 months	90 (20)	17 (13)		11 (11)	6 (16)	
Weight (kg), median (IQR)	7.0 (5.1–9.3)	5.3 (3.9–7.4)	**<0.001**	5.3 (3.6–7.2)	5.3 (4.4–7.7)	0.28
Sex, *n* (%)						
Female	187 (43)	62 (47)	0.40	46 (48)	16 (43)	0.51
Race, *n* (%)						
Caucasian	263 (60)	77 (58)	0.70	52 (54)	25 (68)	0.16
Non-Caucasian	177 (40)	56 (42)		44 (46)	12 (32)	
Ethnicity, *n* (%)						
Hispanic	87 (20)	23 (17)	0.52	14 (15)	9 (24)	0.18
Non-Hispanic	353 (80)	110 (83)		82 (85)	28 (76)	
Insurance, *n* (%)						
Medicaid	171 (39)	56 (42)	0.50	43 (45)	13 (35)	0.31
Other	269 (61)	77 (58)		53 (55)	24 (65)	

NRS = noninvasive respiratory support, IMV = invasive mechanical ventilation, PICU = pediatric intensive care unit, IQR = interquartile range, mo = months; ^†^ Continuous variables are presented as median (interquartile range) and compared using the Wilcoxon rank-sum test; categorical variables are presented as *n* (%) and compared using the chi-square test of independence. Bold highlights *p* values < 0.05.

**Table 2 children-08-01035-t002:** Clinical Characteristics (all groups, *n* = 573).

	NRS (440)	IMV (133)	*p* ^†^	Early IMV (96)	Late IMV (37)	*p* ^†^
Gestational age at delivery						
Median (IQR), week	38 (36–38)	37 (34–38)	**<0.001**	36.5 (34–38)	37 (34.5–38)	0.52
*n* (%)						
Full-term at birth	310 (70)	68 (51)	**<0.001**	48 (50)	18 (49)	0.89
Pre-term at birth	130 (30)	65 (49)		48 (50)	19 (51)	
Late (33 to <37 weeks)	82 (19)	40 (30)	0.56	28 (29)	14 (38)	0.43
Very (28 to <32 weeks)	38 (9)	17 (13)		13 (14)	4 (11)	
Extreme (<28 weeks)	10 (2)	8 (6)		7 (7)	1 (3)	
Pre-existing genetic conditions, *n* (%)	22 (5)	16 (12)	**0.0042**	8 (8)	8 (22)	**0.035**
Pre-existing neurologic conditions, *n* (%)	25 (6)	26 (20)	**<0.001**	18 (19)	8 (22)	0.71

NRS = noninvasive respiratory support, IMV = invasive mechanical ventilation, IQR = interquartile range, wk = weeks; ^†^ Continuous variables are presented as median (interquartile range) and compared using the Wilcoxon rank-sum test; categorical variables are presented as *n* (%) and compared using the chi-square test of independence. Bold highlights *p* values < 0.05.

**Table 3 children-08-01035-t003:** Hospital Course and Pathogen Characteristics (all groups, *n* = 573).

	NRS	IMV	*p* ^†^	Early IMV	Late IMV	*p* ^†^
Intubation, *n* (%)	0	133	–	96 (72)	37 (28)	–
Duration (d), median (IQR)	n/a	7 (5–9)	–	6 (4–9)	7 (6–13.5)	**0.043**
Ventilator-free days, median (IQR)	n/a	21 (19–23)	–	22 (19–24)	21 (15–22)	**0.041**
PICU LOS (d), median (IQR)	2 (1–3)	8 (6–11)	**<0.001**	7 (4.75–10)	10 (7–19.5)	**<0.001**
PIM-III ROM, % (IQR)	1.0 (0.9–1.2)	1.1 (0.9–1.7)	**0.007**	1.1 (0.9–1.8)	1.0 (0.7–1.2)	**0.008**
Vasoactives usage, *n* (%)	0 (0)	16 (12)	**<0.001**	7 (7)	9 (24)	**0.007**
Mortality, *n* (%)	0 (0)	2 (1.5)	**0.01**	2 (2)	0 (0)	0.38
Pathogen *^f^* (#), median (IQR)	1 (1–1)	2 (2–3)	**<0.001**	2 (1–3)	3 (2–4)	**0.007**
Pathogen positive, *n* (%)	398 (97)	131 (98)	0.54	95 (99)	36 (97)	0.48
Single	299 (75)	29 (22)	**<0.001**	25 (26)	4 (11)	0.061
Multiple	99 (25)	102 (78)		70 (74)	32 (89)	
Virus only	391 (98)	36 (27)	**<0.001**	29 (31)	7 (19)	0.48
Bacteria only	0 (0)	4 (3)		3 (3)	1 (3)	
Virus + Bacteria	7 (2)	91 (69)		63 (66)	28 (78)	
Viral count (>1)	94 (24)	29 (22)	0.73	18 (19)	11 (31)	0.15

NRS = noninvasive respiratory support, IMV = invasive mechanical ventilation, IQR = interquartile range, d = days, PICU LOS = pediatric intensive care unit length of stay, PIM-III ROM = pediatric index of mortality-III risk of mortality, # = number; *^f^* Refer to Appendix A for details; ^†^ Continuous variables are presented as median (interquartile range) and compared using the Wilcoxon rank-sum test; categorical variables are presented as *n* (%) and compared using the chi-square test of independence. Bold highlights *p* values < 0.05.

**Table 4 children-08-01035-t004:** Antibiotic Usage, Lab Results (all groups, *n* = 573).

	NRS (440)	IMV (133)	*p* ^†^	Early IMV (96)	Late IMV (37)	*p* ^†^
Total, *n* (%)	126 (29)	125 (94)	**<0.001**	90 (94)	35 (95)	0.85
Duration (d), median (IQR)	7 (3–10)	7 (7–10)	**<0.001**	7 (7–10)	10 (7–14)	**0.017**
Time to initiation (h after intubation), median (IQR)	n/a	2 (0–18)	-	2 (0–10)	8 (0–32)	0.20
WBC (×1000), median (IQR)	n.c.	9.4 (6.8–13)	-	8.9 (6.8–12)	11 (7.7–15)	0.15
% band, median (IQR)	n.c.	0.5 (0–8)	-	1 (0–8)	0 (0–5.3)	0.46
Procalcitonin, median (IQR)	n.c.	1 (0.3–16)	-	1.6 (0.3–16)	0.6 (0.4–17)	0.99

NRS = noninvasive respiratory support, IMV = invasive mechanical ventilation, d = days, IQR = interquartile range, h = hours, n/a = not applicable, n.c. = not checked, WBC = white blood cell; ^†^ Continuous variables are presented as median (interquartile range) or mean (standard deviation) and compared using the Wilcoxon rank-sum test; categorical variables are presented as *n* (%) and compared using the chi-square test of independence. Bold highlights *p* values < 0.05.

## Data Availability

Data will be made available upon reasonable request.

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
