# Peer review of "Risk Factors Associated with Mechanical Ventilation in Critical Bronchiolitis"

_children, 2021, doi:10.3390/children8111035_

Round 1

Reviewer 1 Report

The title and introduction of this paper are very promising; wouldn't it be great if we could distinguish at the time of PICU admission precisely which children with respiratory distress secondary to bronchiolitis are unexpectedly highly likely to require mechanical intubation, because we are currently unable to do so? However, the premise, design, and interpretation are unable to answer this question. Instead, the question they ask "de facto" is - is a computer model able to distinguish between children who did and did not receive mechanical ventilation based on a particular set of criteria (spoiler - only in the discussion do we learn that >50% of those intubated "early" were actually intubated prior to arrival in the PICU). The answer is - not relly. An AUC of 0.716 is not particularly impressive - especially when there is no meaningful attempt to compare this value to that of clinical practice currently - for example, a senior attending on initial assessment. What would have been really interesting here is what is the "added value" that a model can provide. 

Specifically, a few highlights from the major conceptual issues:

Introduction: If viral bronchiolitis is the most common cause of hospital admission, why is it characterized by minor self-limited symptoms? For this study, the relevant context is not that the clinical course is variable, it's that the severity is both variable and unpredictable - and suddenly so. This should be emphasized. Furthermore, you rightfully mention the unique contribution of RSV to the severity of disease in the introduction, but your study design fails to compare/stratify RSV from other pathogens in your primary analysis, nor do you account for days of illness which classically peaks on day 3-5 of illness. 

Methods/Results: The design of the study appears to first and foremost compare the different outcomes by demographic and clinical variables. This is scientifically sound, although not novel. It's not clear why you share intubation data as it not an outcome but rather the definition of your cohort. It does not seem that you are sufficiently powered to compare extubation failure or tracheostomy in the early vs late IMV groups.  Given that some of the NRS received antibiotics, it would be helpful to include time to initiation from admission (to PICU) instead of from intubation. It seems likely that some of those placed on IMV received antibiotics prior to being intubated - and if not, this would be worth addressing. Additionally - the duration of antibiotics is identical (7d) - so a p value of <0.001 is highly unusual and worth commenting if it is not an error. The choice of an AUC for presenting your predictive model is unclear here. Given that you are not assessing the causal relationship, a paper published in 1996 is unlikely to indicate that you have achieved good predictive power, it is difficult to determine the value of the model. Without creating a DAG or using advanced modeling techniques, such as feature selection based on Shapley Values, it is really disingenuous to imply that you are able to determine the most important variables. It seems like the AUC was more of a catchy add-on than a truly meaningful contribution to this study. Sharing the variables and ranges for the logistic regression model would have been more standard and more interpretable. 

Conclusion: It's unclear why you claim that the need for mechanical ventilation was due to bacterial infection, as it is not one of the features you list in your predictive model. Furthermore, given that the NRS group has a negligible bacterial infection rate, it's hard to understand how antibiotic administration could have prevented your outcome of interest (mechanical ventilation).  

In summary, while there is a richness in EMR data here and an important clinical question to be asked - neither the design nor the interpretation is of sufficient quality, novelty, or precision to be a meaningful contribution to the literature. 

Reviewer 2 Report

This is a single-center retrospective study of PICU admissions due to severe infectious bronchiolitis in a tertiary hospital, with aim to look at prognostic risk-factors for invasive mechanical ventilation. The results mostly are not new findings, in terms of risk factors for severe bronchiolitis, but characterize the subgroup at risk of needing ventilation, and also, within this group an insight to the role of bacterial load and possible benefit of antibiotics. The authors also provide a predictive equation of probability for intubation based on risk factors.  

The authors conclude that the patients needing invasive ventilation are a distinct group of patients with risk factors of young age, history of pre-term birth and co-morbidities. They also draw the conclusion, that their more difficult disease course was due to bacterial co-infection and that they would potentially benefit from antibiotics.

Due to retrospective nature, the background data are missing some essential risk factors (smoking exposure, vaccinations and especially RSV-profylaxis) and in over 50% of patients needing mechanical ventilation the exact nature and course of the treatment decisions are lacking. In general, some of the conclusions drawn are not supported fully by the data represented, therefore I have some suggestions and questions:

  • The first impression of the manuscript is somewhat busy and complex, and that a reduction of length would be appropriate. There are 4 tables of results in the manuscript and 6 supplementary tables in addition. Therefore, I would suggest discarding at least the the predictive equation from this manuscript, as it does not add to the current results. Using such an equation clinically would need validation in a prospective clinical trial.
  • In the abstract, you conclude “Patients intubated earlier in their course experienced shorter duration of mechanical ventilation and PICU lengths of stay”, though in the discussion you state that based in the available data, it is not possible to know what was the setting for intubation prior hospital admission (present in >50%)? For the reader this is somewhat confusing ­- please clarify, what do you infer from your data on intubation timing? Please modify your conclusions to reflect the data and results. Also the abstract structure should follow the structure of the whole manuscript.
  • If I understood correctly, no proper airway sampling to look for bacteria was done in the NRS group? NPAs for viruses are reported from both main groups for comparisons, but bacteria only for IMV. In supplementary table 5 the group comparisons (p-value) should be omitted, as there is no real group in NRS bacterial samples.
  • There was a clear difference in antibiotic use frequency between NRS and IMV. Do you have data on if the antibiotics were started before or after starting IMV? What do you think was the rationale of commencing antibiotics i.e is the clinician tempted the commence antibiotics after there is information on bacterial growth from lower airways or is it based on clinical signs pre-emptively? Are there data on lower airway colonization in bronchiolitis patients in general?

I find it hard to follow some of your conclusions in discussion, especially 3rd chapter (note p.11 r 285-287, is the supplementary table numbering right?). Data within IMV subgroup show longer use of antibiotics, and there is a trend towards multiple pathogens in the late IMV group. Also, one third of NRS patients had antibiotics, and expectedly were doing better than IMV patients. You quite strongly conclude at the end of the chapter that “these data strongly associate bacterial coinfections in critical bronchiolitis with need for IMV.” As you do not have comparison data on bacterial infection from the NRS group, I think it is impossible to say what are the confounding factors having effect on the actual outcomes. Therefore, I would suggest clarifying text, and modifying the conclusions to better reflect the data and results regarding the antibiotic use.

Round 2

Reviewer 1 Report

The authors have thoroughly addressed most of the stated concerns. However, there are still a few outstanding issues:

1) Given that they do not typically initiate antibiotics prior to intubation, their concluding sentence ("Early antibiotic administration in these patients has the potential to improve outcomes") while certainly true, is outside the scope of this study - which is instead designed to distinguish which infants require intubation. The authors do not present any evidence that those who were intubated had the opportunity to be given antibiotics at an earlier stage and therefore could have potentially avoid intubation or reduced their risk of death.  

2) The interpretation of an AUC of 0.72 is still inappropriate; interpreting these findings as having "good predictive power" is overstating the capacity of these results and the meaningful limitations of this data set (the study they reference uses "acceptable"). For additional comparison of interpretation of AUC, there are many several recent articles that they might want to consider: (https://pubmed.ncbi.nlm.nih.gov/33262391/, https://pubmed.ncbi.nlm.nih.gov/30274956/, https://pubmed.ncbi.nlm.nih.gov/30795786/)

3) Finally, there is an important semantic issue. They are not generating a prediction model for a future outcome of intubation, in part because many of those including in the study were already intubated at the beginning of their study. Instead, they are (as aptly stated in their title) determining whether certain risk factors are independently associated with having been intubated. As such, the AUC (which is most relevant in assessing predictive modeling and determining thresholds for diagnostic testing, irrespective of the risk factors) does not reassure the reader that the authors have produced a model that can determine which infant will go on to need mechanical ventilation. 

Specifically - at the end of the introduction, the authors appropriately state:

"In this study, we examined patients with acute respiratory failure in the setting of critical bronchiolitis managed on NRS to identify clinical and demographic features that differentiated the NRS group from the IMV group." 

Thus, it is beyond the scope of the study to state in the discussion: 

"By identifying risk factors that have a high index of predicting need for mechanical ventilation, intensivists may better manage these patients with targeted approaches."

I suggest the authors reword their discussion to reflect the intent of the study and the study design - that they have confirmed these risk factors are independently associated with being in the IMV group and further studies should determine early signs for deterioration to require IMV which could be acted upon and potentially result in reduced need for ventilation and thus better outcomes. 

Reviewer 2 Report

I have no further comments and find the changes made to the manuscript acceptable.

Author Response

We wanted to kindly thank reviewer 2 for your acceptance of our revisions.